# Sorbet: A Neuromorphic Hardware-Compatible Transformer-Based Spiking Language Model

Kaiwen Tang [1]    Zhanglu Yan [1]    Weng-Fai Wong [1]

## Abstract

For reasons such as privacy, there are use cases for language models at the edge. This has given rise to small language models targeted for deployment in resource-constrained devices where energy efficiency is critical. Spiking neural networks (SNNs) offer a promising solution due to their energy efficiency, and there are already works on realizing transformer-based models on SNNs. However, key operations like softmax and layer normalization (LN) are difficult to implement on neuromorphic hardware, and many of these early works sidestepped them. To address these challenges, we introduce Sorbet, a transformer-based spiking language model that is more neuromorphic hardware-compatible. Sorbet incorporates a novel shifting-based softmax called PTsoftmax and a Bit Shifting PowerNorm (BSPN), both designed to replace the respective energy-intensive operations. By leveraging knowledge distillation and model quantization, Sorbet achieved a highly compressed binary weight model that maintains competitive performance while achieving $27.16\times$ energy savings compared to BERT. We validate Sorbet through extensive testing on the GLUE benchmark and a series of ablation studies, demonstrating its potential as an energy-efficient solution for language model inference. Our code is publicly available at https://github.com/Kaiwen-Tang/Sorbet

## 1. Introduction

The phenomenal success of large language models (LLMs) (Devlin et al., 2019; Brown et al., 2020; Raffel et al., 2020; Touvron et al., 2023; Lewis et al., 2020; Taylor et al., 2022) has prompted research into distilling *small language models* (SLMs) (Sun et al., 2020; Eldan & Li, 2023; Zhang et al., 2024) from LLMs that can run on the resource-constrained edge devices. Local inference on the device is critical in situations where data privacy is essential or when connectivity to powerful remote computing resources is unfeasible (Sun et al., 2020; Sanh, 2019). The high demand for energy efficiency drives research to simplify SLM inference, making it better suited for devices with low energy consumption while maintaining adequate performance.

In parallel, spiking neural networks (SNNs) have gained significant attention due to their remarkable energy efficiency. SNNs closely mimic biological neural networks and are multiplication-free, resulting in much lower energy consumption compared to artificial neural networks (ANNs). Developing SNNs on neuromorphic hardware significantly improves energy efficiency while preserving impressive performance compared to advanced ANNs (Guo et al., 2023; Shi et al., 2024). For example, the SNN based on the ViT-base architecture (Dosovitskiy et al., 2020) achieves competitive accuracy levels (e.g., up to 81.10% on ImageNet) to its ANN counterpart, but with only one-tenth of energy consumption (Zhou et al., 2024).

However, developing transformer (Vaswani et al., 2017)-based spiking language models is challenging, particularly in encoding spikes and handling the incompatible operations of neuromorphic hardware. Previous works have focused on replacing matrix multiplications with encoding methods, like SpikFormer (Zhou et al., 2024), SpikeBERT (Lv et al., 2023), SpikeLM (Xing et al.), and SpikingBERT (Bal & Sengupta, 2024). However, the transformer architecture also includes *softmax* and *layer normalization* (LN) operations, which are energy-intensive and incompatible with neuromorphic hardware. Models like SpikeLM and SpikingBERT, which target natural language processing (NLP) tasks, still rely on softmax and LN operations, limiting their compatibility with neuromorphic hardware. SpikFormer addresses this issue by adopting features from convolutional networks and batch normalization, avoiding the use of these operations. While effective for vision tasks, their applicability to language tasks, which rely more heavily on operations like LN, remains unproven. Thus, there is no transformer-based

---

[1]School of Computing, National University of Singapore, Singapore, Singapore. Correspondence to: Zhanglu Yan <zlyan@nus.edu.sg>.

*Proceedings of the 42nd International Conference on Machine Learning*, Vancouver, Canada. PMLR 267, 2025. Copyright 2025 by the author(s).

spiking language model that is both hardware-compatible and performs well to date.

*Table 1.* Comparison with other models. The '+' in the NLP row indicates the capability to perform language tasks, while the '+' in the softmax row denotes the inclusion of softmax operations.

|  | BERT | Spikformer | SpikeLM | Sorbet |
|---|---|---|---|---|
| NLP | + | - | + | + |
| softmax | + | - | + | - |
| Norm | LN | BN | LN | BSPN |
| Weight | 32-bits | 32-bits | 1-bit | 1-bit |

In this work, we introduce an SNN-compatible normalization technique called Bit Shifting PowerNorm (BSPN) and a novel shifting-based softmax called Power-of-Two softmax (PTsoftmax). Based on these operations, we propose Sorbet, a novel language model that operates without relying on complex operations. A comparison between Sorbet and previous works is shown in Table 1. To our knowledge, Sorbet is the first model designed for language tasks that overcomes the challenge of deploying transformer-based language models on neuromorphic hardware, which does not support complex operations like division and square roots. In the training process, we apply knowledge distillation techniques to constrain the Sorbet model to binary weights, significantly compressing its size and reducing inference energy consumption. As a result, Sorbet is a highly practical model for resource-constrained devices.

Our tests on the General Language Understanding Evaluation (GLUE) benchmark (Wang et al.) demonstrate that Sorbet maintains stable performance while achieving energy savings of $27.16\times$ compared to BERT and $3.16\times$ compared to SpikeLM. Our contributions are summarized as follows:

- We are the first to explore the neuromorphic hardware-compatible operators in transformer-based models, identifying the challenge of transferring ANNs with transformer structures into SNNs, which lies in operations like softmax and LN. Solving this challenge would complete the last step in enabling SNNs for natural language processing.

- We propose PTsoftmax and BSPN, two operators that replace softmax and layer normalization. These operators rely on bit-shifting instead of expensive operations, making them compatible with neuromorphic hardware and further reducing the model's computational cost.

- We present Sorbet, a transformer-based binary spiking language model derived from BERT. In addition to using these two operators, Sorbet incorporates other design innovations and a refined training process to

achieve full quantization. Sorbet is designed for neuromorphic hardware, enabling energy-efficient inference with comparable performance to ANN counterparts.

## 2. Related Work

**Transformer-based SNNs** Transformed-based SNNs leverage the energy efficiency of SNN architectures. Several transformer-based SNNs have been proposed for computer vision tasks, like Spikformer (Zhou et al.), Spikeformer (Li et al., 2024), Spike-driven Transformer (Yao et al., 2024) and STCA-SNN (Wu et al., 2023). To overcome the limitations of SNNs in handling complex operations, recent developments like Spikformer and Spike-driven Transformer leverage task-specific characteristics and integrate convolutional layers into the architectures. Meanwhile, transformer-based SNNs for NLP tasks have progressed more slowly. The importance of LN in NLP tasks underscores the challenges faced when adapting SNNs for such applications. Recently models include SpikeBERT (Lv et al., 2023), Spiking-BERT (Bal & Sengupta, 2024) and SpikeGPT (Zhu et al.). Among them, SpikeBERT employed even more layer normalization than the original BERT (Devlin et al., 2019), while SpikeGPT and spikingBERT used complicated operations like exponential operation and softmax. In conclusion, none of these works can be practically deployed to neuromorphic hardware.

**Quantized BERT** Model quantization reduces the precision of model weights and activations, such as converting 32-bit floating-point numbers to 8-bit or 4-bit integers. Models like BinaryBERT (Bai et al., 2021) and BiT (Liu et al., 2022) have applied quantization to BERT, achieving notable improvements in model compression and energy efficiency. However, these quantization methods do not align with the requirements of SNNs, as they retain complex operations.

**Simplified Architecture** Several approaches aim to simplify the transformer architecture, such as linear complexity attention mechanisms or hardware-efficient alternatives that offer potential pathways for adaptation (Han et al., 2023; Dao et al., 2022; Lu et al., 2021; Katharopoulos et al., 2020). Additionally, methods have been proposed to simplify computationally-intensive operations within the transformer. For example, I-BERT (Kim et al., 2021) and I-ViT (Li & Gu, 2023) simplify activation functions, normalization functions, and softmax operations with approximation methods. However, these designs are difficult to realize with neuromorphic hardware that do not support multiplications and divisions.

## 3. Preliminary

**Spiking Neural Networks**  SNNs, inspired by biological neural systems, use discrete events (spikes) for communication. Unlike traditional neural networks, SNNs emulate the spike-based communication of neurons, making them more biologically plausible. This mechanism allows efficient processing of temporal data and offers high energy efficiency, making SNNs ideal for applications like robotics, signal processing, and pattern recognition (Kasabov et al., 2013; Kim et al., 2018; Lobov et al., 2020). However, their non-differentiable nature makes training challenging. Current approaches typically involve surrogate gradients or ANN-to-SNN conversion after training an ANN with a similar architecture. In either case, these methods leverage advanced ANN structures to construct analogous SNN models. Surrogate gradients approximate the gradients during back-propagation, enabling SNN training despite their discrete nature. These gradients smooth out non-differentiable spike events. ANN-to-SNN conversion involves transforming a trained ANN model into an SNN by mimicking ANN neuron behavior with the spikes. In Sorbet, we use ANN-to-SNN conversion.

**Spike Neuron Model**  The *integrate and fire* (IF) model is the most popular spike neuron model for generating spike trains (Bu et al.). It offers a simple representation of how SNN neurons accumulate membrane potentials and output spikes. In the IF model, the membrane potential $V$ of a neuron is treated as a capacitor that accumulates the input currents over time, a process that the following differential equation can describe:

$$\frac{dV}{dt} = I_{\text{syn}}(t) \tag{1}$$

Here, $I_{\text{syn}}(t)$ represents the synaptic input current. When the membrane potential $V$ exceeds a certain threshold $\theta$, the neuron generates a spike. In this paper, we adopt an enhanced IF model known as average IF spike generation (ASG) (Yan et al., 2025). One advantage of ASG over the basic IF model is the reduction of memory access energy. Unlike the IF model, which accesses weights at each time step, ASG accumulates input spikes and calculates membrane potential in a single pass, requiring only one access to the weights. The algorithm for ASG can be found in Appendix A Algorithm 4.

**Challenges of Adapting Transformers to SNNs**  While SNNs are recognized for their energy efficiency and explainability, adapting transformer models to spike neurons presents a significant challenge. Spike neurons cannot directly perform essential operations such as multiplication and division, which are fundamental to traditional ANN-based transformers. Consequently, transformer layers in SNNs must be replaced with simpler operations, such as bit-shifting and addition, to ensure compatibility with neuromorphic hardware. This requires substituting ANN transformer layers with SNN-compatible counterparts that avoid complex operations but still maintain the functionality of the original architecture.

## 4. Methods

In this work, we design Sorbet, a spiking language model capable of handling NLP tasks and being practically deployed on neuromorphic hardware platforms. To achieve this, we base the model architecture on transformers due to their proven effectiveness in NLP. As previously discussed in Section 3, neuromorphic hardware does not support standard ANN operations. So we first introduce BSPN (Section 4.1) and PTsoftmax (Section 4.2), which are fully compatible with neuromorphic hardware and provide high-quality results for replacing traditional LN and softmax. We then describe the Sorbet architecture (Section 4.3) and the training process (Section 4.4).

### 4.1. Bit Shifting PowerNorm

Transformer-based language models like BERT typically employ LN. LN calculates the mean and variance across all features for each data point in a layer's input, normalizes the inputs, and applies a learnable scale and shift. Due to hardware limitations discussed earlier, LN cannot be directly adopted in our model. On the other hand, batch normalization (BN) is favored in SNNs because its learnable parameters can be merged into the weights during the inference stage, hence becomes SNN-hardware-friendly. However, the relatively poor performance of BN makes it unsuitable for our model.

To improve BN's performance, PowerNorm (Shen et al., 2020) introduced a relaxed zero mean BN. However, Power-Norm incorporates *root-mean-square layer normalization* (RMSLN):

$$\text{RMSLN}(\mathbf{x}) = \frac{\mathbf{x}}{\sqrt{\frac{1}{n} \sum_{i=1}^{n} x_i^2}} \tag{2}$$

This is too resource-intensive and hence is not fully compatible with neuromorphic hardware.

To address the issue with PowerNorm, we define a novel normalization layer called *Bit Shifting PowerNorm* (BSPN), which replaces RMSLN and avoid complex square and square root operations. We begin by grouping the input. Within each group, denote the vector as $\mathbf{x} \in \mathbb{R}^n$, we calculate the L1 norm.

Then the L1 norm is approximated by the next power of two for hardware efficiency. This allows the "dividing by L1 norm" step to be implemented via bit shifting. The

approximation can be achieved either by taking $\log_2$ and then exponentiating it back, or more efficiently via a lookup table. We then perform the relaxed zero-mean BN as in the PowerNorm. To optimize hardware efficiency, the scaling factor $\frac{\gamma}{\psi}$ can be further quantized to a power of two. The complete BSPN algorithm is outlined in Algorithm 1, where $y \odot X$ is denoted as $[y_1, X_1 :, ..., y_d, X_d :]$.

---

**Algorithm 1** Bit Shifting PowerNorm (BSPN)

1: **Input:** $\mathbf{X} \in \mathbb{R}^{h \times n}$; Number of attention heads $h$;
2: **Output:** $\mathbf{Y}$;
3: **Step 1: Group Scaling**
4: Group channels into $h$ groups.
5: $\mathcal{S} \leftarrow \frac{1}{n} \sum_{i=1}^{n} |\mathbf{X}_i|$
6: $k \leftarrow \lceil \log_2(\mathcal{S}) \rceil$ {Can be done by using a look-up table}
7: $\mathbf{X} \leftarrow \mathbf{X} \gg k$
8: **Step 2: Normalization as PowerNorm**
9: **For Training:**
10: $\psi_B^2 \leftarrow \frac{1}{B} \sum_{i=1}^{B} \mathbf{X}_i^2$
11: $\hat{\mathbf{X}} \leftarrow \frac{\mathbf{X}}{\psi}$
12: $\mathbf{Y} \leftarrow \gamma \odot \hat{\mathbf{X}} + \beta$
13: $\psi^2 \leftarrow \alpha \psi^2 + (1-\alpha)\psi_B{}^2$ {Exponential moving average update for $\psi^2$}
14: **For Inference:**
15: $\mathbf{Y} \leftarrow \gamma \odot \frac{\mathbf{X}}{\psi} + \beta$

---

The two advantages of BSPN are that it uses only SNN-compatible operations and significantly simplifies the computation process. Firstly, like PowerNorm, BSPN incorporates the computation of runtime variance, which is then used during inference. Compared to LN, this eliminates the need for redundant calculations during inference.

Secondly, BSPN avoids RMSLN by using the L1 norm and approximating the divisor as a power of two. These features make BSPN a practical design for SNN models and significantly reduce energy consumption, as shown in Section 4.3 and Section 5.2.

Next, we show that the BSPN is not only deployable on neuromorphic hardware but also possesses desirable properties that make it a suitable replacement for LN in transformer models.

**Definition 4.1.** Let $\Phi \colon \mathbb{R}^n \to \mathbb{R}^n$ be defined by

$$\Phi(X) = \frac{X}{2^{\left\lceil \log_2\left(\mathbf{S}(X)\right) \right\rceil}}, \quad \text{where} \quad \mathbf{S}(X) = \frac{1}{n} \sum_{i=1}^{n} |X_i|. \tag{3}$$

Since bounded gradients are necessary for convergence, we first analyze how the proposed bit-shifting-based step $\Phi(X)$ maintains PowerNorm's gradient boundedness.

**Theorem 4.2** (BSPN Preserves Bounded Gradient). *The*

*loss $L_{\mathrm{PN}}$ under PowerNorm is bounded by a constant, denoted as $C$. We define the BSPN loss by*

$$\mathcal{L}_{\mathrm{BSPN}}(\mathbf{X}) = \mathcal{L}_{\mathrm{PN}}\big(\Phi(\mathbf{X})\big), \tag{4}$$

$\mathcal{L}_{\mathrm{BSPN}}$ *also has a bounded gradient w.r.t. $\mathbf{X}_{:,i}$, specifically*

$$\left\| \frac{\partial \mathcal{L}_{\mathrm{BSPN}}}{\partial \tilde{\mathbf{X}}_{:,i}} \right\| \leq C \quad \text{for all } \tilde{\mathbf{X}}. \tag{5}$$

The detailed proof is provided in Appendix B.1.

To ensure stable training and effective generalization, it is important to control the Lipschitz constant of the loss function. In (Shen et al., 2020), the author proved that applying PowerNorm can lead to a smaller Lipschitz constant of the loss compared to BN under a mild assumption. BSPN exhibits similar behavior, as outlined below:

**Lemma 4.3** (1-Lipschitz Property of $\Phi(X)$). *For any $X, Y \in \mathbb{R}^n$, under mild assumption, we have*

$$\big\| \Phi(X) - \Phi(Y) \big\| \leq \big\| X - Y \big\|. \tag{6}$$

The assumption and detailed proof are provided in the Appendix B.1. For the loss function $\mathcal{L}_{\mathrm{BSPN}}$, for convenience in comparison, we use $\psi_B$ instead of $\psi$. Since $\Phi(X)$ is 1-Lipschitz, BSPN does not increase the overall Lipschitz constant of loss compared to PowerNorm.

**Lemma 4.4** (Effect of BSPN on the Lipschitz Constant of the Loss). *With lemma 4.3 and lemma 2 in (Shen et al., 2020), we further have:*

$$\left\| \frac{\partial \mathcal{L}_{\mathrm{BSPN}}}{\partial \mathbf{X}_{:,i}} \right\|^2 \leq \left\| \frac{\partial \mathcal{L}_{\mathrm{PN}}}{\partial \mathbf{X}_{:,i}} \right\|^2 \leq \frac{\gamma_i^2}{(\psi_B)_i^2} \left\| \frac{\partial \mathcal{L}_{\mathrm{BN}}}{\partial \mathbf{X}_{\cdot,i}} \right\|^2. \tag{7}$$

From (Shen et al., 2020), the empirical result supports that $\frac{\gamma_i}{(\psi_B)_i}$ is typically smaller than 1. Therefore, the Lipschitz constant of $\mathcal{L}_{\mathrm{BSPN}}$ is also smaller than that of $\mathcal{L}_{\mathrm{BN}}$ in practice. These properties of BSPN imply that it enhances training stability by preventing gradient explosion or vanishing, ensuring controlled updates. This makes BSPN a robust and efficient alternative to LN.

### 4.2. Power-of-Two Softmax

In transformer-based models, the softmax function plays a crucial role, especially in the attention mechanisms where it is used to calculate the distribution of attention weights across different inputs. For a vector $\mathbf{z} = [z_1, z_2, ..., z_n]$, softmax can be calculated as follows:

$$\mathrm{softmax}(z_i) = \frac{e^{z_i}}{\sum_{j=1}^{n} e^{z_j}} \tag{8}$$

Due to the complexity of the exponential and division operations involved in the softmax function, it is too sophisticated for neuromorphic hardware, making direct utilization of softmax in SNNs impractical. We propose a softmax alternative aligned with SNN computational conventions, enabling a more streamlined attention mechanism suitable for SNNs.

To approximate the softmax function, we first replace the exponential operation with powers of two, thus defining a base-2 softmax:

$$\text{Base-2 softmax}(z_i) = \frac{2^{z_i}}{\sum_{j=1}^{n} 2^{z_j}} \quad (9)$$

Since the base-2 softmax still involves division, we further approximate $\sum_{j=1}^{n} 2^{z_j}$ as the nearest power of two $\tilde{Z}$:

$$k = \left[\!\!\left[ \left( \log_2 \left( \sum_{j=1}^{n} 2^{\lceil z_j \rceil} \right) \right) \right]\!\!\right], \quad \tilde{Z} = 2^k \quad (10)$$

Here, $k$ is the integer logarithm base 2 of the sum, rounded up, ensuring $\tilde{Z}$ is the nearest power of two approximating the softmax denominator. To facilitate bit-shifting, we also round up $z_i$. Our proposed purely power-of-two softmax (PTsoftmax) can then be defined as:

$$\text{PTsoftmax}(z_i) = \frac{2^{\lceil z_i \rceil}}{\sum_{j=1}^{n} 2^{\lceil z_j \rceil}} \approx 2^{\lceil z_i \rceil - k} \quad (11)$$

Given $z_i$ and $k$, the operation $2^{z_i - k}$ can be efficiently computed via a left shift operation. The complete PTsoftmax computation is detailed in Algorithm 2.

---

**Algorithm 2** Power-of-two Softmax (PTsoftmax)

---

1: **Input:** Attention scores matrix $\mathcal{S} \in \mathbb{R}^{b \times s \times h}$
2: **Output:** Attention probabilities matrix $\mathcal{P} \in \mathbb{R}^{b \times s \times h}$
3: $\mathcal{S}' \leftarrow \lceil \mathcal{S} \rceil - \max(\lceil \mathcal{S} \rceil)$ {Normalize $\mathcal{S}$}
4: $\mathcal{A} \leftarrow 2^{\mathcal{S}'}$
5: $\mathcal{Z} \leftarrow \sum(2^{\mathcal{S}'})$ {Compute the sum of exponentials}
6: $k \leftarrow [\![\log_2(\mathcal{Z})]\!]$
7: $\mathcal{P} \leftarrow \mathcal{A} \gg k$ {Use right shift to divide by $\tilde{\mathcal{Z}} = 2^k$}
8: **return** $\mathcal{P}$

---

To analyze the approximation error of PTsoftmax relative to the original softmax, we followed (Zhang et al., 2022), and used the generalized base-$\beta$ softmax function, $F_\beta(x_i)$, defined as:

$$F_\beta(x_i) = \frac{\beta^{x_i}}{\sum_{j=1}^{N} \beta^{x_j}} \quad (12)$$

Here, the traditional softmax corresponds to $F_e(x_i)$, and the base-2 softmax is $F_2(x_i)$. According to (Zhang et al., 2022), both $F_e(x_i)$ and $F_2(x_i)$ exhibit similar behavior and can be

used interchangeably. In particular, $F_2(x_i)$ is well suited to representing a probability distribution in $(0, 1]$ such that the probabilities sum to 1. We then examine the approximation between $F_2(x_i)$ and PTsoftmax.

**Lemma 4.5.** *For all $i \in 0, 1, ..., n$, we have $\frac{1}{2\sqrt{2}} F_2(x_i) \leq$ PTsoftmax$(x_i) \leq 2\sqrt{2} F_2(x_i)$.*

A detailed proof is given in Appendix B.2. Lemma 4.5 shows that the approximation error of PTsoftmax remains within a constant factor of the traditional softmax, confirming its practical applicability. Unlike classic softmax, PTsoftmax does not strictly sum to 1, but theoretical analysis and experimental results (Section 5.3) indicate that this discrepancy has a minor impact on performance.

### 4.3. Sorbet Architecture

We now present Sorbet, our spiking language model. At a high level, Sorbet resembles a standard transformer design but is customized for the SNN platform by replacing various layers with SNN-compatible ones, including BSPN (Section 4.1), PTsoftmax (Section 4.2), and ReLU, equipped with additional spike neurons and avoiding multiplication with accumulation operations. Figure 1 shows the architecture of Sorbet along with the current ANN design (e.g., BERT).

Each activation is encoded using spike neurons to generate spike trains. Our spiking self-attention mechanism is then defined as:

$$\begin{aligned} \text{SpikingAttn}(x) = \\ \mathcal{SN}(\text{PTsoftmax}(\alpha * \mathcal{SN}(Q)K^T))V \end{aligned} \quad (13)$$

where $Q, K, V$ are derived from a quantized binary-weight linear function, and $\alpha$ is a constant that can be merged into the weights. $\mathcal{SN}$ denotes the spike neuron used to generate spike trains. In Sorbet, at the position of matrix multiplication in the ANN, one of the multiplicands is encoded into a spike train. Matrix multiplications involving $\mathcal{SN}$ are simplified to accumulations over the indices corresponding to spike events (1s in the spike train).

In sub-layers originally defined as *LayerNorm*($x$ + *Sublayer*($x$)), we instead use *BSPN*($\mathcal{SN}(x)$ + *BinaryLayer*($\mathcal{SN}(x)$)). To preserve performance after quantization, we also adjust the training process. Notably, existing neuromorphic chips such as Loihi, IBM TrueNorth, and NeuroGrid (Davies et al., 2018; Akopyan et al., 2015; Benjamin et al., 2014) already support spiking and bit-shifting operations, making them well suited for these innovations.

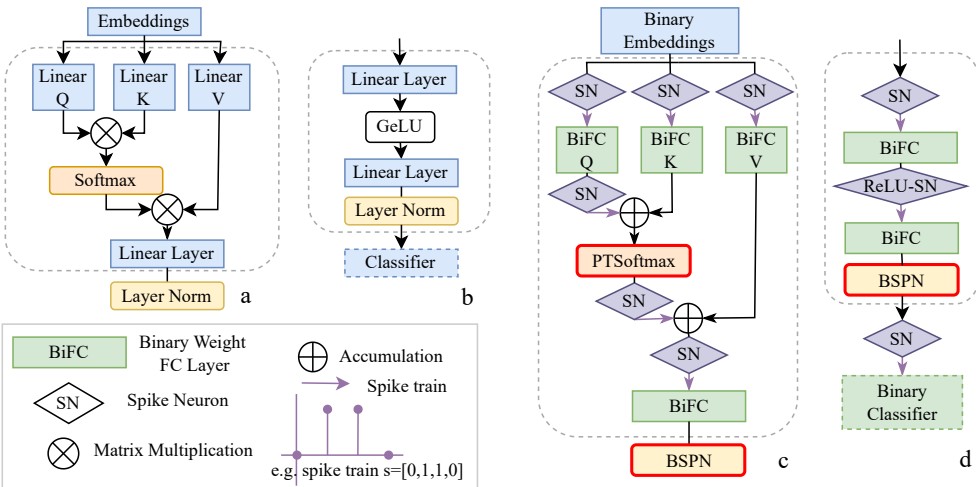

*Figure 1.* Comparison of the architecture of BERT and Sorbet. (a) The multi-head self-attention block of BERT; (b) The feed-forward network of BERT; (c) The spiking multi-head self-attention of Sorbet; (d) The spiking feed-forward network of Sorbet. All the outputs of $\mathcal{SN}$ are spike trains to ensure Sorbet is multiplication-free. The red-bordered box highlights our proposed operations.

## 4.4. Training Process

Firstly, to enhance the energy efficiency of our model and enable the encoding of all activations into spike trains, we quantize all weights to 1 bit and activations to 4 bits. For model quantization, (Liu et al., 2022) introduced an elastic binarization function with a scale factor $\alpha$ and threshold $\beta$:

$$X_B^i = \alpha \hat{X}_B^i = \alpha \lfloor \text{Clip}(\frac{X_R^i - \beta}{\alpha}, 0, 1) \rfloor \qquad (14)$$

However, dividing the input by $\alpha$ is impractical for SNN inference. Hence, following the same rationale as PTsoftmax, we approximate $\alpha$ with the nearest power of two, $Z = 2^{k_\alpha}$. Accordingly, the elastic binarization function becomes:

$$X_B^i = \left\lfloor \text{Clip}\left((X_R^i - \beta) \gg k_\alpha, 0, 1\right) \right\rfloor \ll k_\alpha \qquad (15)$$

Inspired by (Liu et al., 2022), we adopt a hybrid training strategy that combines standard knowledge distillation with the distillation of intermediate activations. The overall loss function is $L = L_{\text{logits}} + L_{\text{reps}}$, where $L_{\text{logits}}$ employs the Kullback-Leibler (KL) divergence to facilitate learning from the teacher model to the student model while $L_{\text{reps}}$ is used to accelerate convergence and improve transfer and generalization capabilities (Aguilar et al., 2020). Concretely,

$$L_{\text{logits}} = \text{KL}(p, q), \ L_{\text{reps}} = \sum_i \|r_i^s - r_i^t\|^2 \qquad (16)$$

where $p$ denotes the output distribution of the teacher model, and $q$ represents the output of the student model. $r_i^s$ and $r_i^t$ are the corresponding transformer block output activations

from the student and teacher models, respectively. The backpropagation can be calculated as:

$$
\begin{aligned}
\frac{\partial L}{\partial w} &= \sum_i \left( \frac{\partial L}{\partial p_i}\frac{\partial p_i}{\partial w} + \frac{\partial L}{\partial q_i}\frac{\partial q_i}{\partial w} + \frac{\partial L}{\partial r_i^s}\frac{\partial r_i^s}{\partial w} + \frac{\partial L}{\partial r_i^t}\frac{\partial r_i^t}{\partial w} \right) \\
&= \sum_i \left( \left( \log\left(\frac{p_i}{q_i}\right) + 1 \right)\frac{\partial p_i}{\partial w} - \frac{p_i}{q_i}\frac{\partial q_i}{\partial w} \right) \\
&\quad + \sum_i \left( 2(r_i^s - r_i^t)\frac{\partial r_i^s}{\partial w} - 2(r_i^s - r_i^t)\frac{\partial r_i^t}{\partial w} \right)
\end{aligned}
$$
$$\qquad (17)$$

Then we integrate BSPN and PTsoftmax step by step into the model. For each step, we perform model distillation. Finally, the quantized model with these energy-efficient components is transformed into the Sorbet model by passing it through spiking neurons. This entire procedure is summarized in Algorithm 3.

---

**Algorithm 3** Multi-step distillation

1: **Input:** Full-precision model $M_0$, dataset $\mathcal{D}$
2: **Output:** Sorbet $\mathcal{S}$
3: $M_1 \leftarrow$ Quantize($M_0$) {Quantize $M_0$ to 1-bit weight 4-bit activation}
4: $M_2 \leftarrow M_1$ with PTsoftmax replacing softmax
5: $M_3 \leftarrow M_2$ with BSPN replacing LN
6: **for** $i = 1 \rightarrow 3$ **do**
7: $\quad M_{\text{teacher}} \leftarrow M_{i-1}, M_{\text{student}} \leftarrow M_i$
8: $\quad$ ModelDistill($M_{\text{student}}, M_{\text{teacher}}, \mathcal{D}$)
9: **end for**
10: Convert $M_3$ to SNN and obtain Sorbet $\mathcal{S}$
11: **return** $\mathcal{S}$

---

The distillation process yields a quantized model with higher accuracy. Furthermore, as shown in Table 3, our proposed PTsoftmax and BSPN can also be applied and trained directly.

# 5. Result

In this section, we present the performance of Sorbet on seven datasets from the GLUE benchmark, a standard evaluation suite used by plenty of language models. Because SNNs are relatively limited when applied to NLP tasks, there are few existing SNNs evaluated on GLUE, so we compare Sorbet against both SNN and quantized ANN baselines. Our experiments use BERT-base as the initial teacher model for distillation. We also provide a comprehensive analysis of the model's energy and power efficiency. All experiments were conducted on three Nvidia RTX A100 GPUs, each with 80GB of memory. The number of timesteps used for all results in this section is 16.

## 5.1. Comparing with the Baseline

Table 2 shows Sorbet's performance on the GLUE benchmark, indicating that it maintains competitive results overall. It achieves state-of-the-art performance on four datasets and attains comparable results on the rest. Compared to binary neural networks such as BiT, Sorbet has the same model size and comparable performance but offers better energy efficiency and compatibility with neuromorphic hardware.

Additionally, we also evaluated two existing SNNs, namely, SpikeLM and SpikingBERT, on GLUE with 1-bit weight quantization. Although they claim to be SNN architectures with spike-generation methods, both depend heavily on LN and softmax operations, which are incompatible with SNN hardware. By contrast, Sorbet avoids these constraints and is consequently more suitable for neuromorphic devices.

## 5.2. Energy Saving Analysis

Sorbet achieves substantial energy efficiency in three ways. Firstly, SNN reduces overall energy consumption due to its event-driven nature, activating neurons only when necessary. Secondly, PTsoftmax and BSPN replace traditional functions with lower-cost operations, further reducing energy use. Finally, the use of model quantization further reduces computational load and power requirements.

Compared to ANNs, the key energy-saving advantage of SNNs lies in replacing multiplications with additions. The numbers of addition needed in Sorbet($N_{\text{Sorbet}}$) to substitute matrix multiplication in BERT($N_{\text{BERT}}$) can be expressed as:

$$N_{\text{Sorbet}} = T \cdot r \cdot N_{\text{BERT}} \tag{18}$$

where $T$ is the timestep and $r$ is the spike rate. From Eq. 18,

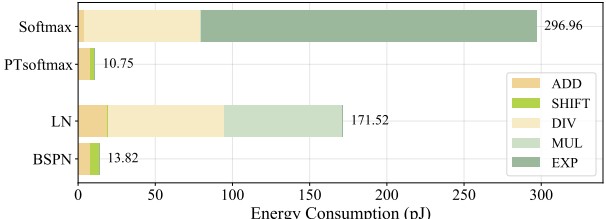

*Figure 2.* Energy cost of different operations. Each value represents a single execution with an input dimension of 128. Based on 45nm technology, a FIX8 division requires 0.59 pJ, whereas a bit-shift operation requires only 0.024 pJ.

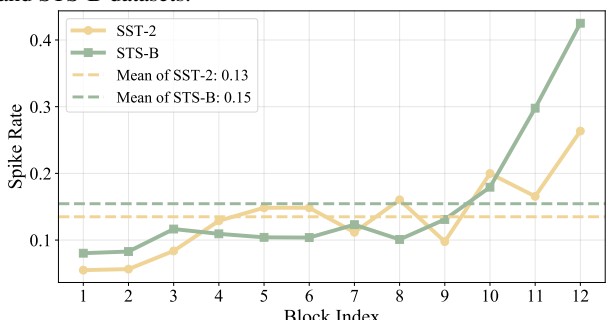

*Figure 3.* Spike firing rate for the output of each block on SST-2 and STS-B datasets.

$N_{\text{Sorbet}}$ is highly related to the spike rate $r$. For example, on the SST-2 and STS-B datasets, the observed average spike rates are 0.13 and 0.15, respectively, as illustrated in Figure 3. Notably, spike rates can be higher when using symmetric quantization. With these observed spike rates, we calculated Sorbet's energy consumption and compared it with other baselines in Table 4. Sorbet reduces energy consumption by $27.16\times$ compared to BERT and by $3.16\times$ compared to SpikeLM.

The simplification operations we propose also substantially reduce the computational load of these functions. Using the reported energy costs of DIV, EXP, and SHIFT operations (You et al., 2020; Nadh et al., 2023; Wu et al., 2019), we calculated the energy consumption of our PTsoftmax and BSPN compared to the standard softmax and LN in Figure 2. The results show that PTsoftmax achieves approximately $27.62\times$ better energy efficiency than conventional softmax, while BSPN achieves $12.4\times$ better energy efficiency than LN. Detailed calculations are provided in Appendix E.

## 5.3. Ablation Study

To evaluate the contribution of our proposed components, we conducted a series of ablation experiments. Specifically, we focused on the effectiveness of the PTsoftmax and BSPN modules. We conducted two ablation studies to evaluate the impact of our proposed modifications. First, we replaced

| Model | Size | QQP | MNLI-m | SST-2 | QNLI | RTE | MRPC | STS-B |
|---|---|---|---|---|---|---|---|---|
| BERT$_{base}$ (Devlin et al., 2019) | 418M | 91.3 | 84.7 | 93.3 | 91.7 | 72.6 | 88.2 | 89.4 |
| DistilBERT (Sanh, 2019) | 207M | 88.5 | 82.2 | 91.3 | 89.2 | 59.9 | 87.5 | 86.9 |
| TinyBERT$_6$ (Jiao et al., 2020) | 207M | - | 84.6 | 93.1 | 90.4 | 70.0 | 87.3 | 83.7 |
| Q2BERT (Zhang et al., 2020) | 43.0M | 67.0 | 47.2 | 80.6 | 61.3 | 52.7 | 68.4 | 4.4 |
| BiT (Liu et al., 2022) | 13.4M | 82.9 | 77.1 | 87.7 | 85.7 | 58.8 | 79.7 | 71.1 |
| SpikingFormer (Zhou et al., 2023) | * | 83.8 | 67.8 | 82.7 | 74.6 | 58.8 | 74.0 | 72.3 |
| SpikingBERT (Bal & Sengupta, 2024) | 50M | 86.8 | 78.1 | 88.2 | 85.2 | 66.1 | 79.2 | 82.2 |
| SpikeLM (Xing et al.) | * | 87.9 | 76.0 | 86.5 | 84.9 | 65.3 | 78.7 | 84.3 |
| 1-bit SpikingBERT (Bal & Sengupta, 2024) | * | 83.8 | 75.4 | 86.7 | 80.5 | - | 75.8 | - |
| 1-bit SpikeLM (Xing et al.) | * | 87.2 | 74.9 | 86.6 | 84.5 | 65.7 | 78.9 | 83.9 |
| Sorbet ‡ | 13.4M | 83.4 | 75.8 | 89.6 | 84.6 | 59.2 | 78.4 | 73.6 |
| Sorbet | 13.4M | 86.5 | 77.3 | 90.4 | 86.1 | 60.3 | 79.9 | 78.1 |

*Table 2.* Comparison with the baseline on the GLUE benchmark. * denotes that the size is not reported in the original work. We report Spearman correlations for the STS-B dataset and accuracy for all other datasets. ‡ denotes further quantizing BSPN's scaling factor $\frac{\gamma}{\psi}$ to a power of two.

*Table 3.* Ablation study on PTsoftmax and BSPN in full precision ANNs

| Model | QQP | MNLI-m | SST-2 | QNLI | RTE | MRPC | STS-B | Avg. |
|---|---|---|---|---|---|---|---|---|
| BERT-softmax-LN | 91.3 | 84.7 | 93.3 | 91.7 | 72.6 | 88.2 | 89.4 | 87.3 |
| BERT-PTsoftmax-LN | 90.8 | 83.9 | 91.4 | 90.8 | 71.5 | 85.3 | 87.6 | 85.9 |
| BERT-PTsoftmax-BSPN | 89.7 | 80.9 | 91.7 | 87.4 | 69.0 | 81.9 | 84.4 | 83.6 |

*Table 4.* Energy cost comparison with various baselines. Sorbet is first evaluated on the STS-B dataset, while * denotes usage on the SST-2 dataset.

| Model | BERT | LIF-BERT | SpikeLM |
|---|---|---|---|
| FP32 | 51.41 | 7.98 | 3.98 |
| FP16 | 15.21 | 3.55 | 1.77 |
| | **Ours** | | |
| 1-Bit | | 0.65 / 0.56 * | |

*Table 5.* Ablation study on the impact of PTsoftmax and BSPN. $\delta$ is the accuracy drop compared to a model using Softmax and LayerNorm at the same precision. All models use 1-bit weights; 'Bits' in this table denotes the activation quantization bit-width.

| PTsoftmax | BSPN | Accuracy (%) | $\delta$ |
|---|---|---|---|
| | **Bits = 4** | | |
| × | × | 91.5 | - |
| ✓ | × | 90.8 | 0.7 |
| × | ✓ | 91.2 | 0.3 |
| ✓ | ✓ | 90.9 | 0.6 |
| | **Bit = 1** | | |
| × | × | 81.2 | - |
| ✓ | × | 80.0 | 1.2 |
| × | ✓ | 79.9 | 1.3 |
| ✓ | ✓ | 79.8 | 1.4 |

the softmax and LN components in the full-precision BERT model with our PTsoftmax and BSPN, respectively. The performance results of this replacement are detailed in Table 3. The impact caused by the two components is equivalent to the model performance. Compared to our main result on Sorbet in Table 2, the accuracy drop from full precision BERT to Sorbet is mainly caused by the quantization of weight and spike generation process, not by the replacement of softmax and normalization. Exploring more accurate ways to perform model quantization and spike generation can be a potential future work.

Second, we tested the effectiveness of our components in highly quantized BERT models on the SST-2 datasets. The results are presented in Table 5. Our proposed PTsoftmax and BSPN can maintain a good performance on full precision and highly quantized models. We also performed

ablation study on quantization level and timestep with the conversion loss on SST-2 dataset in Appendix D.2.

## 6. Discussion

Although we do not have access to physical neuromorphic chips, to demonstrate the neuromorphic hardware compatibility of our proposed model, we have implemented and validated the PTsoftmax and BSPN layers using the Lava

framework, targeting Intel's Loihi architecture. We further synthesized PTsoftmax and BSPN layers in Verilog for a commercial 22 nm process to estimate power consumption as in Appendix E. While these methods demonstrate substantial energy-efficiency gains, they cannot fully capture real hardware effects. Future work will prioritize deployment on actual neuromorphic platforms to obtain empirical power and latency measurements, validate our simulation fidelity, and guide hardware-aware refinements.

## 7. Conclusion

In this paper, we presented Sorbet, the first fully neuromorphic hardware-compatible, transformer-based spiking language model. Sorbet addresses the challenge of adapting transformer-based models for energy-efficient computation by replacing traditional energy-intensive operations like softmax and layer normalization with our novel PTsoftmax and BSPN. This challenge has been largely overlooked in previous works in this field. Furthermore, by using techniques such as knowledge distillation and model quantization, we achieved a highly compressed binary weight model, further optimizing the model for real-world deployment on neuromorphic hardware. When evaluated on the GLUE benchmark, Sorbet achieved performance comparable to state-of-the-art models with substantial energy savings.

At the time of writing this paper, DeepSeek has become a global phenomenon. DeepSeek-V3 (DeepSeek-AI, 2024) uses quantization to FP8 on Nvidia GPUs and standard floating point softmax and normalization operations, while DeepSeek-R1 (DeepSeek-AI, 2025) innovates in the training and distillation processes to achieve energy and resource-efficient models. We believe that the techniques of Sorbet can be adapted to make them even more so, especially for inference on customized embedded architectures. This shall be the focus of our future work.

## Acknowledgment

This research is supported by the Ministry of Education, Singapore, under the Academic Research Fund Tier 1 (FY2024).

## Impact Statement

This paper presents work whose goal is to advance the field of machine learning. There are many potential societal consequences of our work, none of which we feel must be specifically highlighted here.

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

## A. Spike Generation

We provide the detailed spike generation method we adopted in algorithm 4, where $w_{ij}^l$ is the weight of layer $l$ from neuron $i$ to neuron $j$, $b_i^l$ is the bias of the neuron $i$ in layer $l$, $s_i^l$ is the input spike train of the neuron $i$, and $T$ is the time window size.

---

**Algorithm 4** Average IF model

---

1: **Input:** Weight $w_{ij}^l$, bias $b_i^l$, input spike train $s_i^l$, threshold $\theta$, membrane potential $U_i^l(t)$ at timestep $t$, time window size $T$;
2: **Output:** Spike train $s_j$;
3: $U_i^l(t) \leftarrow \sum_j (w_{ij}^l \cdot s_i^l(t) + b_i^l)$
4: $A_i^l \leftarrow \sum_{t=1}^{T} U_i^l(t)/T$
5: $s_i^l(0) \leftarrow 0, V_i^l(0) \leftarrow 0$;
6: **for** $t \leftarrow 1$ to $T$ **do**
7:    $V_i^l(t) \leftarrow V_i^l(t-1) + A_i^l$
8:    $s_j^{l+1}(t) \leftarrow \begin{cases} 1, & V_i^l(t) \geq \theta \\ 0, & \text{otherwise} \end{cases}$;
9:    $V_i^l(t) \leftarrow \begin{cases} V_i^l(t) - \theta, & V_i^l(t) \geq \theta \\ V_i^l(t), & \text{otherwise} \end{cases}$.
10: **end for**
11: **return** $s_j^{l+1}$ {Output the ASG spike train of neuron $i$}

---

## B. Theoretical Results

### B.1. Proof of BSPN

**Assumption B.1** (Power-of-Two Condition on Normalization Input). For a Transformer-based SNN, the input to the normalization layer $X$ satisfies

$$\lceil \log_2 (\mathbf{S}(\mathbf{X})) \rceil \geq 0,$$

where

$$\mathbf{S}(\mathbf{X}) = \frac{1}{n} \sum_{i=1}^{n} |X_i|.$$

This condition is mild and consistent with our empirical observations in Transformer-based SNNs. The empirical results are shown in Appendix D.1.

Based on these results, we have the following proof of Theorem 4.2 and Lemma 4.3.

*Proof of Theorem 4.2.* Since $L_{\text{BSPN}}(\mathbf{X}) = L_{\text{PN}}(\Phi(\mathbf{X}))$, based on chain rule, we have:

$$\frac{\partial L_{\text{BSPN}}(\mathbf{X})}{\partial \mathbf{X}} = \frac{\partial L_{\text{PN}}(\Phi(\mathbf{X}))}{\partial \Phi(\mathbf{X})} \cdot \frac{\partial \Phi(\mathbf{X})}{\partial \mathbf{X}}. \tag{19}$$

$$\frac{\partial \Phi(\mathbf{X})}{\partial \mathbf{X}} = \frac{\frac{\partial \mathbf{X}}{\partial \mathbf{X}} \cdot 2^{\lceil \log_2 (\mathbf{S}(\mathbf{X})) \rceil} - \mathbf{X} \cdot \frac{\partial}{\partial \mathbf{X}} 2^{\lceil \log_2 (\mathbf{S}(\mathbf{X})) \rceil}}{\left(2^{\lceil \log_2 (\mathbf{S}(\mathbf{X})) \rceil}\right)^2}. \tag{20}$$

Since

$$\frac{\partial \mathbf{X}}{\partial \mathbf{X}} = I, \quad \text{and} \quad \frac{\partial}{\partial \mathbf{X}} 2^{\lceil \log_2 (\mathbf{S}(\mathbf{X})) \rceil} = 0 \quad \text{(almost everywhere)}, \tag{21}$$

we obtain

$$\frac{\partial \Phi(\mathbf{X})}{\partial \mathbf{X}} = \frac{I_n}{2^{\lceil \log_2 (\mathbf{S}(\mathbf{X})) \rceil}}. \tag{22}$$

Thus, the gradient of $L_{\mathrm{BSPN}}(\mathbf{X})$ can be expressed as:

$$\frac{\partial L_{\mathrm{BSPN}}(\mathbf{X})}{\partial \mathbf{X}} = \frac{I}{2^{\lceil \log_2(\mathbf{S}(\mathbf{X})) \rceil}} \cdot \frac{\partial L_{\mathrm{PN}}(\Phi(\mathbf{X}))}{\partial \Phi(\mathbf{X})}. \tag{23}$$

As the gradient of any $\mathbf{X}$ is bounded for $L_{\mathrm{PN}}$(see section 4.2 in (Shen et al., 2020)):

$$\left\| \frac{\partial L_{\mathrm{BSPN}}(\mathbf{X})}{\partial \mathbf{X}} \right\|_2 \leq \frac{1}{2^{\lceil \log_2(\mathbf{S}(\mathbf{X})) \rceil}} \cdot C \tag{24}$$

Now, under the mild Assumption B.1, we have:

$$\lceil \log_2(\mathbf{S}(\mathbf{X})) \rceil \geq 0. \tag{25}$$

Thus,

$$2^{\lceil \log_2(\mathbf{S}(\mathbf{X})) \rceil} \geq 1 \quad \Rightarrow \quad \frac{1}{2^{\lceil \log_2(\mathbf{S}(\mathbf{X})) \rceil}} \leq 1. \tag{26}$$

Therefore, we conclude:

$$\left\| \frac{\partial L_{\mathrm{BSPN}}(\mathbf{X})}{\partial \mathbf{X}} \right\|_2 \leq C. \tag{27}$$

This establishes the gradient bound. $\qquad\square$

*Proof of Lemma 4.3.* Let $k_X = \lceil \log_2\big(\mathbf{S}(X)\big) \rceil$ and $k_Y = \lceil \log_2\big(\mathbf{S}(Y)\big) \rceil$, under Assumption B.1, we have $k_X, k_Y \geq 0$, hence

$$\frac{1}{2^{k_X}}, \frac{1}{2^{k_Y}} \leq 1. \tag{28}$$

Therefore, we have:

$$\big\| \Phi(X) - \Phi(Y) \big\| = \big\| \frac{X}{2^{k_X}} - \frac{Y}{2^{k_Y}} \big\| = \big\| \frac{2^{k_Y} X - 2^{k_X} Y}{2^{k_Y} 2^{k_X}} \big\| \leq \big\| 2^{k_Y} X - 2^{k_X} Y \big\| \tag{29}$$

$$\leq \max(2^{k_Y}, 2^{k_X}) \big\| X - Y \big\| \leq \big\| X - Y \big\|. \tag{30}$$

$\qquad\square$

## B.2. Proof of PTsoftmax

In this section, we provide detailed proof of the approximation error of PTsoftmax and base-2 softmax.

*Proof of Lemma 4.5.* Let $a_i = F_2(x_i)$, $b_i = F_2(\lceil x_i \rceil)$, $c_i = 2^{\lceil x_i \rceil - k}$, where $k = \left\lceil \log_2 \left( \sum_{j=1}^{n} 2^{x_j} \right) \right\rceil$, We claim the following inequalities:

1. $\frac{1}{2} b_i \leq a_i < 2 b_i$.
2. $\frac{1}{\sqrt{2}} c_i \leq b_i \leq \sqrt{2} c_i$.

**Proof of 1:**   Since $x_i \leq \lceil x_i \rceil < x_i + 1$, we consider the worst-case scenarios. For the right side, the worst case occurs when $x_i = \lfloor x_i \rfloor + \epsilon$, where $\epsilon$ is an arbitrarily small value, no greater than 1, and $\lceil x_j \rceil = x_j$ for all $j$. Denote $\lfloor x_i \rfloor$ as $n_i \in \mathbb{Z}$. Then $\lceil x_i \rceil = n_i + 1$, so we have:

$$b_i = \frac{2^{n_i + 1}}{\sum_{j=1}^{N} 2^{x_j} - 2^{n_i + \epsilon} + 2^{n_i + 1}} = \frac{2 \cdot 2^{n_i}}{\sum_{j=1}^{N} 2^{x_j} - \epsilon + 1} \tag{31}$$

$$\geq \frac{2 \cdot 2^{n_i}}{\sum_{j=1}^{N} 2^{x_j}} \geq 2 \cdot \frac{2^{n_i + \epsilon}}{\sum_{j=1}^{N} 2^{x_j}} = 2 a_i. \tag{32}$$

For the left side, the worst case occurs when $\lceil x_i \rceil = x_i$ and $x_j = n_j + \epsilon$ for some $j$, where $n_j = \lfloor x_j \rfloor$. Then $\lceil x_j \rceil = n_j + 1$, and we have:

$$b_i = \frac{2^{x_i}}{\sum_{j=1}^{N} 2^{n_j+1} - 2^{x_i+1} + 2^{x_i}} = \frac{2^{x_i}}{2 \cdot \sum_{j=1}^{N} 2^{n_j} - 2^{x_i}} \tag{33}$$

$$< \frac{2^{x_i}}{2 \cdot \sum_{j=1}^{N} 2^{n_j}} \leq \frac{2^{x_i}}{2 \cdot \sum_{j=1}^{N} 2^{n_j+\epsilon}} = \frac{1}{2} a_i \tag{34}$$

**Proof of 2:** Given that $k = \left\lceil \log_2 \left( \sum_{j=1}^{n} 2^{x_j} \right) \right\rceil$, we have the following bounds:

$$k - 0.5 \leq \log_2 \left( \sum_{j=1}^{n} 2^{\lceil x_j \rceil} \right) \leq k + 0.5. \tag{35}$$

Thus, the ratio

$$\frac{b_i}{c_i} = \frac{2^k}{\sum_{j=1}^{N} 2^{\lceil x_j \rceil}}. \tag{36}$$

satisfies

$$\frac{1}{\sqrt{2}} = \frac{2^k}{2^{k+0.5}} \leq \frac{b_i}{c_i} \leq \frac{2^k}{2^{k-0.5}} = \sqrt{2} \tag{37}$$

$\square$

## C. Evaluation Benchmark

We evaluate our Sorbet on 7 distinct datasets in the GLUE benchmark as follows:

- **MNLI**: The MNLI (Multi-Genre Natural Language Inference Corpus) is involved in natural language inference tasks. It consists of a collection of sentence pairs annotated for textual entailment through crowdsourcing.

- **QQP**: The QQP (Quora Question Pairs) pertains to tasks involving similarity and paraphrase identification, focusing on pairs of questions from the community Q&A website, Quora. The primary objective of this task is to ascertain whether a pair of questions are semantically equivalent.

- **QNLI**: The QNLI (Question-answering Natural Language Inference) is a task in natural language inference. QNLI is derived from another dataset, the Stanford Question Answering Dataset (SQuAD 1.0), which is a question-paragraph pair question-answering dataset where the paragraphs are sourced from Wikipedia.

- **SST-2**: The SST-2 (Stanford Sentiment Treebank) is a single-sentence classification task that involves sentences from movie reviews and their sentiment annotations by humans. This task requires classifying the sentiment of a given sentence into positive and negative sentiment.

- **STS-B**: The STS-B (Semantic Textual Similarity Benchmark) comprises a collection of sentence pairs extracted from sources such as news headlines, video titles, image captions, and natural language inference data. It is a regression task.

- **RTE**: The RTE (Recognizing Textual Entailment datasets) are from natural language inference tasks. It consolidates datasets from a series of annual textual entailment challenges, with data samples constructed from news sources and Wikipedia.

- **MRPC**: The MRPC (Microsoft Research Paraphrase Corpus) is involved in similarity and paraphrase tasks. It consists of sentence pairs automatically extracted from online news sources, with human annotations to determine if the sentences are semantically equivalent. The categories are not balanced, with 68% of the samples being positive instances.

## D. Extra Results

### D.1. Empirical Results

We present the empirical results for $\lceil \log_2(\mathbf{S}(\mathbf{X})) \rceil$ collected from different layers of Sorbet in Figure 4. These results confirm that Assumption B.1 holds, as all values are greater than zero.

*Figure 4.* Distribution of $\mathbf{S}(\mathbf{X})$ measured from various Sorbet layers. The strictly positive values support the assumption made in Assumption B.1.

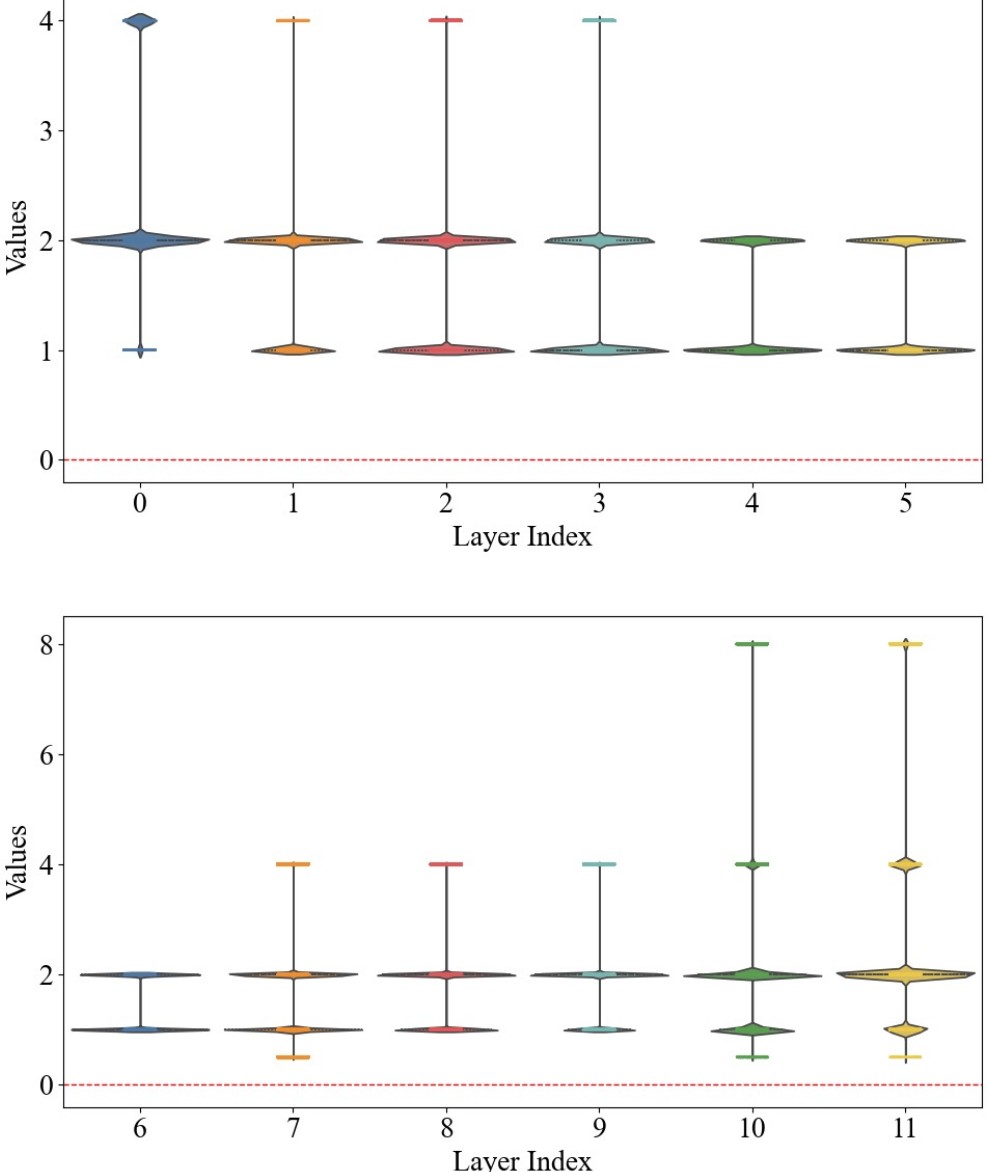

### D.2. Ablation on Quantization Levels

We provide additional results for different quantization levels and timesteps as a supplement to our ablation study on the SST-2 dataset, shown in Table 6.

*Table 6.* Accuracy comparison between quantized ANNs and SNNs under different activation bits and timesteps. $\delta$ is the accuracy drop when converting quantized ANNs into SNNs.

| Activation Bits / Timestep | | 1 / 2 | 2 / 4 | 3 / 8 | 4 / 16 |
|---|---|---|---|---|---|
| Accuracy(%) | Quantized ANN | 79.8 | 87.9 | 90.1 | 90.9 |
| | SNN | 78.7 | 87.4 | 89.3 | 90.4 |
| $\delta$ | | 1.1 | 0.5 | 0.8 | 0.5 |

# E. Energy Analysis

## E.1. Computational cost for whole model

For assessment of the energy consumption of the whole model, we calculate based on the energy usage of multiplications and additions. For sorbet, the weights are binary while the timestep is 16, so the maximum accumulation can be represented with 1-bit FIX and 4-bit FIX. We tested on 22nm technology 0.01215pJ. As all the baselines are under a 45nm process technology, to make it a fair comparison, we approximate 1 accumulation in Sorbet as $E_{ACC} = 0.0243$pJ.

Note that for original ANN BERT, we use $E_{MAC} = 4.6$pJ (Horowitz, 2014) for 1 multiplication. Thus, with the spike firing rate $r$ and the total number of time steps $T$, the energy cost of Sorbet can be calculated as:

$$E_{\text{Sorbet}} = T \cdot r \cdot E_{\text{BERT}} \cdot \frac{E_{ACC}}{E_{MAC}}. \tag{38}$$

## E.2. Computational cost for operations

In Table 7, we list for a input with dimension $d$, the needed operations for different softmax and normalization:

*Table 7.* Computational cost comparison of the PTsoftmax and BSPN with their equivalents.

| | + | - | $\times$ | $\div$ | $e^x$ | $x^2$ | $\sqrt{x}$ | $\gg$ | LUT |
|---|---|---|---|---|---|---|---|---|---|
| softmax | $n-1$ | - | - | $n$ | $n$ | - | - | - | - |
| PTsoftmax | $n-1$ | $n$ | - | - | - | - | - | $n$ | 1 |
| LayerNorm | $3n-2$ | $2n$ | $2n$ | $n+2$ | - | $n$ | 1 | - | - |
| BSPN | $2n-1$ | - | - | - | - | - | - | $2n$ | 1 |

For our energy calculation in Figure 2, we count the major operations as 0.03pJ for addition, 0.2pJ for multiplication, 0.024pJ for shifting (You et al., 2020), 0.59pJ for division (Nadh et al., 2023) and 1.7pJ for exponential (Wu et al., 2019). To make it a fair comparison, we use 8-bit integers for all these operations. For the results in Figure 2, we take $d = 128$. For example, the energy cost of applying softmax once can be calculated as:

$$E_{\text{softmax}} = d \cdot (E_{\text{ADD}} + E_{\text{DIV}} + E_{\text{EXP}}) = 128 \times (0.03 + 0.59 + 1.7) = 296.96pJ. \tag{39}$$

