# OpenReview forum: "Sorbet: A Neuromorphic Hardware-Compatible Transformer-Based Spiking Language Model"
_ICML.cc/2025/Conference — ICML 2025 poster_

### Official Review · Reviewer_MRPY · 2025-03-11

**Overall Recommendation:** 4

**Summary:**

This paper argues that current Transformer-based SNN language models are difficult to deploy on neuromorphic chips due to the presence of softmax and layer normalization operations. To address this challenge, the authors propose Sorbet, a model that is more compatible with neuromorphic hardware. Sorbet is based on the concept of shifting and integrates novel PTsoftmax and BitShifting-based PowerNorm algorithms, which avoid complex operations like division and exponentiation. Additionally, the paper introduces techniques such as knowledge distillation and model quantization to enhance model performance and energy efficiency. Sorbet achieves comparable performance to other SNN language models on the GLUE benchmark.

**Claims And Evidence:**

Yes. The claims made in the submission supported by clear and convincing evidence.

**Essential References Not Discussed:**

No.

**Experimental Designs Or Analyses:**

I checked the soundness/validty of the experimantal designs and analyses. The main results, energy saving analysis and ablation study are necessary due to the claim of energy efficiency and competitive performance.
However, I believe that the experiments and explanations should be further supplemented. Overall, the proposed method integrates module design, quantization, distillation, and spiking neurons, making the training process relatively complex. Therefore, the authors should clearly describe the experimental setup of these components and provide sufficient ablation studies; otherwise, readers may find it confusing. I would like to raise the following questions regarding the experiments and hope the authors can address them and revise the paper accordingly. This is important for me to reconsider the rating.
1. The experiments are based on the BERT model, which consists of two stages: pre-training and fine-tuning. However, the paper does not explicitly mention these two critical stages, even in Supplementary A's Algorithm 4. This makes it unclear whether Algorithm 4 entirely pertains to the pre-training phase. For example, when is the spiking neuron introduced—right after pre-training, or only after fine-tuning? Additionally, does the fine-tuning phase still involve knowledge distillation?
2. In Line 306(left column), who is the teacher model? Is it BERT_base?
3. In Algorithm 4, why is knowledge distillation conducted in three steps? Is there any literature supporting this approach? What are the hyperparameters (e.g., learning rate, epochs) for each step of distillation? Unfortunately, I could not find any details on experimental hyperparameters in the paper.
4. Since Sorbet is ultimately an SNN model with activations taking only 0 or 1, why is 4-bit activation quantization performed first? Would it be possible to skip activation quantization and directly convert the model to an SNN? If activation quantization is necessary, please provide ablation studies to support its significance.
5. In Table 2, two versions of Sorbet are listed. What are the differences between them in terms of weight and activation quantization? In Line 344, does "a power of two" refer to the 1-bit weight quantization mentioned in Line 260 (right column)? If so, what is the relationship between the Sorbet models in Lines 340 and 341?
6. In Table 4, does "Bits" refer to weight quantization or activation quantization? In Line 423 (left column), the authors state that "the accuracy drop from full-precision BERT to Sorbet is mainly caused by the quantization of weight and spike generation process, not by the replacement of softmax and normalization." This may suggest that Table 4 refers to weight quantization. However, given that the model also uses 4-bit activation quantization, what is its impact on performance? Additionally, how does the spike generation process affect the model’s accuracy? I could not find any ablation study on introducing spiking neurons.
7. In Line 134 (left column), the authors adopt a novel ASG model instead of the traditional IF model, arguing that "ASG accumulates input spikes and calculates membrane potential in a single pass, requiring only one access to the weights." However, based on Algorithm 3, it seems that the IF model could also load weights in a single pass by computing the summed inputs for t from 1 to T, similar to Step 3, but without averaging as done in Step 4. The authors' reasoning here might be misleading. Furthermore, I would like to see ablation studies on how different spiking neuron models affect performance.

**Methods And Evaluation Criteria:**

Yes. The proposed Sorbet and evaluation criteria make sense for the problem.

**Other Comments Or Suggestions:**

1. In the section of Spiking Neural Networks within the "Preliminary" chapter, the author mentions terms such as "surrogate gradients" and "ANN-to-SNN" without providing explanations or references, which may lead to confusion. Furthermore, which specific method does the SNN in this paper employ?
2. In Line 810, The energy cost are from (). Maybe the content in () is overlooked.
3. When I checked the code in Suplementary material E, I found the repository is expired.

**Other Strengths And Weaknesses:**

N/A

**Questions For Authors:**

1. What is the relationship between k and \psi_B in Algorithm 1?
2. What is the proportion of energy consumption reduction contributed by efficient methods including BSPN, PTSoftmax, quantization, and spiking neurons? Which one plays the primary role?
3. The seven questions of experiment mentioned in Experimental Designs Or Analyses, which are important.

**Relation To Broader Scientific Literature:**

A key computational characteristic of SNNs is sparse operations, primarily based on addition, which has been demonstrated in models like SpikFormer and Spike-driven Transformer. Recently, there has been a surge of work on Transformer-based SNN language models, focusing on the critical challenge of handling the Attention mechanism. Some approaches, like SpikeBERT, attempt to discard the Softmax operation similarly to SpikFormer; however, doing so in language tasks often leads to poor performance unless distillation techniques are applied. On the other hand, SpikeLM retains the Softmax operation. This work explores an alternative approach—constructing an Attention mechanism using a shift-based method while preserving a Softmax-like structure, making it a valuable contribution to the field.

**Theoretical Claims:**

Yes. The theoretical claims include 1. bit-shifting-based step maintains PowerNorm's gradient boundness 2. the approximation error of PTsoftmax remains within a constant factor of the traditional softmax. In my opinion, both of them are clear and there are no obvious errors.
Moreover, I am concerned about the credibility of Assumption B.1. Line 651 states, "In practical scenarios, the activations in Transformer-based SNNs are typically non-trivial, ensuring that their L1 norm remains above 1." However, in Figure 1 (c) and (d), the input X of  of the BSPN module is not binary (0-1). I hope the authors can provide a more detailed explanation, whether through theoretical analysis or experimental evidence.

---

> ### Author Rebuttal · Authors · 2025-04-01
>
> Thank you for your thorough review and valuable suggestions. We hope our response below can address your concerns.
>
> ---
> > Q1 & 2: Pretraining, fine-tuning, and distillation
>
> We use a pre-trained and fine-tuned BERT from HuggingFace as a starting point. We then applied distillation and quantization techniques, as in Algorithm 4, without further fine-tuning. After these, the weights are directly converted into a corresponding SNN. This part is not the main contribution of Sorbet. **In our Sorbet experiments, the teacher model refers to BERT_base**. We will clarify this in the updated version.
>
> ---
> > Q3: Multi-step distillation(MSD)
>
> The MSD method we use is based on BiT[1]. Section 5.5 of that paper showed that MSD yields better performance. MSD is also adopted in other recent works[2]. We performed distillation in three steps because we applied three major modifications to the model before SNN conversion.
>
> [1] Liu et al. Bit: Robustly binarized multi-distilled transformer. Neurips, 2022.
>
> [2] Han et al. Amd: Automatic multi-step distillation of large-scale vision models.ECCV, 2024.
>
> We will add the hyperparameters in appendix. We included them in the same link of our code.
>
> ---
> > Q4: Skipping activation quantization
>
> Activation quantization is essential for SNNs from ANN-to-SNN conversion to reduce timestep and maintain accuracies. A quantized ANN can resemble an SNN, allowing the model to learn how to represent quantized values in the equivalent SNN. Recent studies have also proposed lossless conversion methods through activation quantization[3].
>
> [3] Shen et al. Are conventional snns really efficient? A perspective from network quantization. CVPR 2024.
>
> As an ablation study, we convert a full-precision BERT into an SNN with T=16. The model achieved only 50.92 accuracy on the SST-2 dataset, indicating a failure to represent the data.
>
> ---
> > Q5: Two versions of Sorbet in Table 2
>
> The two versions of Sorbet only differ in their **quantization strategies for BSPN**. For the Sorbet$\ddagger$(line 340), we quantize BSPN’s weight to powers of two. These weights refer to the scaling factor $\frac{\gamma}{\psi}$ in line 15 of Algorithm 1. We will clarify this in our revised manuscript.
>
> ---
> > Q6: 'Bits' in Table 4
>
> In Table 4, 'Bits' refers to the quantization bit-width of activations. For 'Bits=4', weights are 1-bit and activations are 4-bit. For 'Bits=1', both weights and activations are 1-bit.
>
> As quantizing the activations is necessary to realize the spiking neurons, the energy-saving impact of quantization and spiking cannot be evaluated separately. However, the loss due to spiking neurons after quantization can be measured as follows:
>
> | Activation Bits/Timestep | 1/2  | 2/4  | 3/8  | 4/16 |
> | --- | ---- | ---- | ---- | ---- |
> | Acc (Quantized ANN)  | 79.8 | 87.9 | 90.1 | 90.9 |
> | Acc (SNN)  | 78.7 | 87.4 | 89.3 | 90.4 |
> | Loss  | 1.1  | 0.5  | 0.8  | 0.5  |
>
> ---
> > Q7: Why ASG & ablation for spike neuron
>
> The traditional IF model loads weights, computes membrane potentials, and generates spikes at every timestep, thus requiring multiple reads of the weights. Algorithmically, the IF model can be optimized by first summing inputs across all $T$ timesteps to reduce repeated weight accesses. However, this optimization would require extra storage for membrane potentials at each timestep. ASG avoids storing intermediate data. Furthermore, we conducted an ablation study on the SST-2 dataset. For T=2, ASG achieves 78.7% accuracy, while IF achieves 57.1%. For T=4, ASG reaches 87.4%, while IF is at 79.7%. These results demonstrate that ASG outperforms the IF model, further justifying our choice of ASG.
>
> ---
> > Q8: Other suggestions
>
> We will add the explanation in preliminary: Surrogate gradients approximate the gradients during backpropagation, enabling SNN training despite their discrete nature. These gradients smooth out non-differentiable spike events. ANN-to-SNN conversion involves transforming a trained ANN model into an SNN by mimicking ANN neuron behavior with the spike. **In Sorbet, we use ANN-to-SNN conversion.**
>
> We will fix the missing reference in the appendix. The extended repository and code are now available at the same link as in the manuscript.
>
> ---
> > Q9: k and \psi_B in Algorithm 1
>
> Thank you for pointing out. There is a mistake in line 10, Algorithm 1:
>
> $$\sigma_B^2 \gets \frac{1}{B}\sum_{i=1}^{B}\mathbf{X}_i^2$$
>
> should be
>
> $$\psi_B^2 \gets \frac{1}{B}\sum_{i=1}^{B}\mathbf{X}_i^2$$
> We hope that clarifies.
>
> ---
> > Q10: Energy saving proportion
>
> SNNs are energy efficient because of their sparsity and low-bitwidth operations. Quantization and spiking neurons contribute to more than 99% of the energy-saving. Even though BSPN and PTSoftmax by themselves do not consume much energy, without them, it is not possible to realize true SNNs and the benefits that SNNs bring.
>
> ---
> > Q11: Explanation of Assumption B.1
>
> The plot for the distribution of the input of normalization can be found in the code repository.

---

> > ### Comment · Reviewer_MRPY · 2025-04-05
> >
> > Thank you very much for your patient reply. Although some of the responses lack detail, I understand the challenge of addressing so many questions within the character limit. Overall, the work presented in this paper is solid and could help stimulate more discussion in the field of SNNs. I have decided to raise my score. Since the answers to these questions are somewhat brief and scattered, I sincerely hope the authors can provide a clearer description of the experimental procedures, necessary explanations and references in future revisions to enhance readability.

---

> > > ### Author Response · Authors · 2025-04-09
> > >
> > > Thank you very much for your feedback on our work and for taking the time to provide such constructive comments. We would like to make additional clarifications on several key issues. Due to time and space constraints here, we will incorporate your suggestions and provide a more detailed and systematic description of the experimental procedures in our revised manuscript.
> > >
> > > ---
> > > > Q1 & 2 Extension: Pretraining, fine-tuning, and distillation
> > >
> > > As we mentioned in our previous response, our starting point is already fine-tuned with the target dataset. We have added the download links in the code repository. To boost the energy efficiency of our model and enable the encoding of all activations into spike trains, we quantize all weights to 1-bit and activations to 4-bits. This step adopts the model distillation method detailed in [1]. With the incorporation of BSPN and PTsoftmax, the revised model is treated as a student model. After distillation, the weights will be fixed and transferred to an SNN directly.
> > >
> > > [1] Liu Z, Oguz B, Pappu A, et al. Bit: Robustly binarized multi-distilled transformer. NeurIPS, 2022.
> > >
> > > ---
> > > > Q3 Extension: Training hyperparameters
> > >
> > > We will add the training details in our appendix to provide a clear path for reproducing our work. Specifically, the parameters for each step vary from task to task as follows:
> > > | **Dataset** | **Epochs** | **Max Seq Length** | **Batch Size** | **Learning Rate** |
> > > |-------------|----------------------|--------------------|----------------|-------------------|
> > > | MNLI        | 100                  | 128                | 120            | 1e-5              |
> > > | MRPC        | 100                  | 128                | 40             | 1e-6              |
> > > | SST-2       | 200                  | 64                 | 180            | 1e-6              |
> > > | STS-B       | 200                  | 128                | 30             | 5e-7              |
> > > | QQP         | 150                  | 128                | 100            | 1e-5              |
> > > | QNLI        | 150                  | 128                | 80             | 1e-6              |
> > > | RTE         | 100                  | 128                | 10             | 5e-6              |
> > >
> > > ---
> > > > Q4 Extension: Activation Quantization
> > >
> > > A quantized ANN can be deemed equivalent to an SNN because by quantizing activations in an ANN, continuous outputs are transformed into discrete signals that closely resemble the threshold-based spiking mechanism inherent in SNNs. In SNNs, neurons fire only when their membrane potential exceeds a specific threshold, producing binary outputs. Similarly, quantization in ANNs acts as a filter, suppressing sub-threshold activations and preserving only those above a defined cutoff, thus effectively mimicking the discrete, event-driven behavior of spiking neurons.
> > >
> > > We can also offer other references to show that performing activation quantization before conversion is mainstream, namely [2] and [3].
> > >
> > > [2] Bu T, Fang W, Ding J, et al. Optimal ANN-SNN Conversion for High-accuracy and Ultra-low-latency Spiking Neural Networks. ICLR 2023.
> > >
> > > [3] Hu Y, Zheng Q, Jiang X, et al. Fast-SNN: Fast spiking neural network by converting quantized ANN. IEEE TPAMI, 2023.
> > >
> > > ---
> > > > Q8 Extension: Preliminary for SNN
> > >
> > > As mentioned in the paper, we either directly train an SNN or convert an ANN into an SNN. We adopted the latter in this paper because it has been shown to converge faster as well as produce better accuracies when compared to the former [4, 5]. Direct training of SNNs requires surrogate gradients that not only introduce additional design and tuning challenges but also have issues related to gradient approximation and delay, making the training process more complex and prone to getting stuck in local optima. Secondly, by converting a well-trained ANN to an SNN, we can theoretically maintain performance levels comparable to those of the original ANN, whereas training SNNs directly with surrogate gradients may result in more significant performance fluctuations due to approximation errors and instability.
> > >
> > > [4] Jiang H, Anumasa S, De Masi G, et al. A unified optimization framework of ANN-SNN conversion: towards optimal mapping from activation values to firing rates. ICML, 2023.
> > >
> > > [5] Huang Z, Shi X, Hao Z, et al. Towards High-performance Spiking Transformers from ANN to SNN Conversion. ACM MM, 2024.

---

### Official Review · Reviewer_ck8m · 2025-03-14

**Overall Recommendation:** 3

**Summary:**

The authors propose Sorbet: A transformer-based spiking language model optimized for neuromorphic hardware, enhancing energy efficiency while maintaining strong performance. It introduces BitShifting-based PowerNorm (BSPN) for normalization and Power-of-Two softmax (PTsoftmax) as a hardware-friendly alternative to softmax. Through binary weight quantization via knowledge distillation, Sorbet achieves 27.16x energy savings over BERT and 3.16x compared to SpikeLM while remaining competitive on the GLUE benchmark.

**Claims And Evidence:**

Yes, the claims appear well-supported by the experimental results and visualizations.

**Essential References Not Discussed:**

I believe the authors have covered a range of relevant references related to SNNs, transformer-based models, and quantization techniques.

**Experimental Designs Or Analyses:**

Yes, the experiments look sound. Like, the paper evaluates Sorbet against multiple SOTA methods and includes ablation studies to highlight the effectiveness of the PTsoftmax and BSPN modules.

**Methods And Evaluation Criteria:**

Yes

**Other Comments Or Suggestions:**

Correct the orientation of captions of Figs. 2 and 3.

**Other Strengths And Weaknesses:**

Strengths:-

1.) The paper is well-written and articulated.

2.) The authors propose the first transformer-based spiking language model that removes softmax and Layer Normalization.

3.) PTsoftmax and BSPN are proposed to replace softmax and layer normalization, using bit-shifting instead of costly operations, making Sorbet more efficient for neuromorphic hardware.

4.) They propose Sorbet, a binary spiking language model derived from BERT, integrating full quantization and design refinements to enable low-power, high-efficiency inference comparable to ANN models.

5.) A broad range of experiments (multiple datasets, models, baselines, and ablations) are conducted, and their experimental results demonstrate the effectiveness of Sorbet over existing methods. It achieves 27.16x energy savings over BERT and 3.16x over SpikeLM while maintaining stable performance on the GLUE benchmark.


Weaknesses:-

1.) The authors did the theoretical energy estimation of SNN architectures rather than empirical validation. The efficacy of their approach (PTsoftmax and BSPN) would have been more realized if the authors had deployed the converted SNN on neuromorphic hardware —such as Intel Loihi, TrueNorth, or BrainChip Akida —and provided measured power consumption data.

2.) While SNNs offer low power consumption, they inherently introduce additional latency due to spike processing over T timesteps. Although the proposed Sobret is made quantized and more efficient for neuromorphic accelerator, it still results in higher latency compared to ANNs. For real-world applications where inference time is critical, it remains unclear how SNNs can effectively address this challenge. A quantitative latency analysis, particularly in time-sensitive scenarios, would have been valuable. Additionally, evaluating the inference time could have provided deeper insights into the Sobret’s practical impact.

3.)  Hardware Dependency: The practical deployment of SNN models rely on neuromorphic hardware, which seems limited in accessibility and widespread adoption. The authors should have addressed this limitation in their study.

4.)  A discussion of the potential limitations of the Sorbet approach would enhance the paper's credibility.

**Questions For Authors:**

I would like all points under weaknesses to be addressed.

**Relation To Broader Scientific Literature:**

The paper contributes within the domains of small language models for edge devices, SNNs for energy efficiency, and transformer-based SNNs for NLP tasks. It also builds on research in quantized BERT models and simplified transformer architectures. The authors acknowledge prior work in these areas and present Sorbet as a solution to key limitations in existing approaches, particularly for NLP tasks on neuromorphic hardware.

**Theoretical Claims:**

Yes, like theorems have proof in the appendix section. I read them and seemed justified enough.

---

> ### Author Rebuttal · Authors · 2025-04-01
>
> Thank you for your thorough review and valuable suggestions. Below are our responses, which we hope will address your concerns.
>
> ---
>
> > Q1: Deployment of SNNs on neuromorphic hardware
>
> We have evaluated the hardware compatibility with the Lava framework to simulate Loihi chip. However, the platform does not provide the energy cost. So we independently implemented our designs of the two functions in Verilog and tested power consumption using a commercial 22nm FD-SOI technology process and reported in the manuscript. The results indicate that our proposed functions PTsoftmax and BSPN achieve approximately 27.63x and 12.4x better energy efficiency compared to conventional implementations.
>
> ---
>
> > Q2: Latency and inference time
>
> A direct comparison between ANNs and SNNs on the issue of latency is not straightforward. It will depend on model and hardware parameters, as well as circuit implementation and optimization. Also, below certain thresholds required by the application, it becomes a non-issue.
>
> All things being equal, the latency of SNNs increases with the timestep. We experimented with setting the Sorbet timestep to 2 and achieved an accuracy of 78.7 on the SST-2 dataset, demonstrating the robustness of Sorbet.
>
> Compared to an ANN, even using the same model size, the hardware circuits of an SNN would be simpler than an ANN. This would translate to a higher frequency. While it may take a few cycles for an SNN to produce what an ANN can do in a single cycle, because of the higher frequency, the SNN may still perform better on the end-to-end latency, or at least be able to satisfy the application's requirements.
>
> ---
>
> > Q3: Hardware Dependency
>
> While SNNs are still not widely used, many companies are actively exploring neuromorphic hardware such as Intel, IBM, BrainChip, Qualcomm, and so on, reflecting the strong interest and potential in this emerging field. We believe that once SNN models can achieve comparable performance of advanced ANNs and be efficiently deployed on neuromorphic hardware using techniques such as what is proposed here, these chips will be more widely adopted as viable alternatives to overcome the power and latency limitations of traditional digital computing.
>
> ---
>
> > Q4: Limitation
>
> We will add a discussion section on the limitations of Sorbet. Sorbet is designed for SNN deployment specifically on edge devices. Sorbet optimizes the model with constrained computational resources. Hence, it probably will not outperform larger, more complex models in environments without resource constraints.
>
> ---
>
> > Q5: Figure Caption
>
> Thanks for pointing this out, we will fix it in our paper. We have extended the repository, and the code is available at the same link provided in the manuscript.

---

### Official Review · Reviewer_GqJG · 2025-03-14

**Overall Recommendation:** 3

**Summary:**

This paper proposed a Spiking Transformer language model, named Sorbet, designed for neuromorphic hardware. Sorbet introduces two approximations, PTsoftmax and BSPN, to replace traditional softmax and layer-wise normalisation.  The aim of them is to make the model neuromorphic compatible and energy-efficient. PTsoftmax replaces softmax's exponential and division operations with bit-shifting.  BSPN approximates the L1 norm to the nearest power of two, avoiding square and square root operations.  Sorbet also integrates binary weight quantization with knowledge distillation to further reduce the model's computational cost.  On the GLUE benchmark, Sorbet can achieve competitive performance, yet with much reduced energy consumption.

**Claims And Evidence:**

One of the key claims of this study is a neuromorphic hardware compatible model.  Hence evidence provided should have a hardware focus, which is not the case in the paper.  Can the model be evaluated on actual chips, such as Loihi, IBM TrueNorth, or NeuroGrid? It is more important than showing high efficiency.

The choice for baseline methods needs a bit update to substantiate the claim of high performance high efficiency of Sorbet. There are some efficient models like DistilBERT [1] or TinyBERT [2] which are designed for edge devices. How would Sorbet compare in terms of energy efficiency and performance trade-offs against these efficient models? In addition,  recent spiking transformers should be considered in the comparison, such as spikeBERT [3].  Also ANN to SNN conversion methods should be considered as well such as QCFS [4].  SpikeGPT is mentioned in the paper, but not included in the comparison.

[1] Sanh et al., “DistilBERT, a distilled version of BERT: smaller, faster, cheaper and lighter”, NeurIPS 2019.
[2] Jiao et al., “TinyBERT: Distilling BERT for Natural Language Understanding”, EMNLP 2020.
[3] Lv et al., Spikebert: A language spikformer learned from bert with knowledge distillation, AAAI 2024
[4] Bu, et al. Optimal ann-snn conversion for high-accuracy and ultra-low-latency spiking neural networks, ICLR 2022.

In Table 2, the performance of SpikingBERT is list as: 83.8 75.4 86.7 80.5 - 75.8 - on the seven GLUE datasets.  However the original paper reported better results: 86.82 78.10 88.19 85.20 66.06 79.17/85.15 82.20/81.90 respectively.  Why is there such discrepancy?  The results from SpikeLM are also different to that on Xing's paper.

Another point is the time step.  It is not mentioned in the main results on Table 2.  Does Sorbet fix it to 16?  If so, what would happen for a different timestep?

**Essential References Not Discussed:**

See the claims section.  More references should be added.

**Experimental Designs Or Analyses:**

The paper’s experimental design and analyses appear to be sound and comprehensive.  The comparison is performed using the well established GLUE benchmark. Several baselines, e.g., BERT, SpikeLM, and other quantized models, are involved.  The energy saving analysis and theoretical proof are included in the paper.  In addition, ablation studies are performed to show the contributions of PTsoftmax and BSPN.

**Methods And Evaluation Criteria:**

The proposed approximation methods, PTsoftmax and BSPN, are interesting and promising.  In this sense, this study may lead to a practical way to replace energy-intensive operations in Transformers, which could be a critical step for enabling language models on neuromorphic hardware.

The evaluation is performed on the GLUE benchmark, comparing Sorbet's performance with several baselines, including quantised models and other SNN-based language models.  This study is solid in this regard.

**Other Comments Or Suggestions:**

SpikeLM by Xing et al., "SpikeLM: Towards general spike-driven language modeling via elastic bi-spiking mechanisms" is published in ICML 2024, not just an arXiv article.

Figure 3. is not necessary, a simple table would be clear and more concise.

Two latex problems in Table 1 caption, '+'  --> `+'

Line 370 Where -> where

**Other Strengths And Weaknesses:**

See above

**Questions For Authors:**

How to use Sorbet as a multimodal LLM?  That is an important extension to be considered.

"27.16× energy savings compared to BERT" was mentioned multiple times.  How was it calculated?

Also see questions in the above sections.

**Relation To Broader Scientific Literature:**

This work could contribute towards high-performance transformers designed for neuromorphic hardware, offering theory and practice of efficient neural network design.

**Theoretical Claims:**

It is nice to see theoretical proofs in the paper.  The analysis confirms that BSPN maintains bounded gradients, making it a robust and efficient alternative to LN.  Also, although PTsoftmax does not strictly sum to 1, the analysis shows that this discrepancy has a minor impact on performance.

In addition, the ablation studies confirm that the use of PTsoftmax and BSPN introduces minimal performance degradation.

---

> ### Author Rebuttal · Authors · 2025-03-31
>
> Thank you for your thorough review and valuable suggestions. Below are our responses, which we hope will address your concerns.
>
> ---
> > Q1: Hardware focus and evaluation on actual chips
>
> We appreciate your suggestions. To demonstrate the neuromorphic hardware compatibility of our proposed model, we have implemented and validated the PTsoftmax and BSPN layers using the Lava framework, targeting Intel's Loihi architecture. We created an AbstractProcess along with its corresponding PyLoihiProcessModel, deployed within Lava's simulation environment Loihi1SimCfg.
>
> We do not have access to physical neuromorphic chips. However, beyond the simulation, we implemented our design in Verilog and evaluated its power consumption using a commercial 22nm FD-SOI technology process. The results shows that PTsoftmax and BSPN achieve approximately $27.63\times$ and $12.4\times$ better energy efficiency than conventional operations. We will include the result into our final version.
>
> ---
>
> > Q2: Compare with more baselines
>
> Thank you for the suggestion, we have added the following comparison and will include them in our result section. In terms of FLOPs, TinyBERT\_6 and DistilBERT\_4 reduce energy by 2.0× and 3.0×, respectively, compared to BERT, while Sorbet achieves a remarkable 27.16× reduction:
>
> | Model      | Size(MB) | QQP  | MNLI-m | SST-2 | QNLI | RTE  | MRPC | STS-B |
> | ---------- | -------- | ---- | ------ | ----- | ---- | ---- | ---- | ----- |
> | BERT_base  | 418      | 91.3 | 84.7   | 93.3  | 91.7 | 72.6 | 88.2 | 89.4  |
> | DistilBERT | 207      | 88.5 | 82.2   | 91.3  | 89.2 | 59.9 | 87.5 | 86.9  |
> | TinyBERT_6 | 207      | -    | 84.6   | 93.1  | 90.4 | 70.0 | 87.3 | 83.7  |
> | Sorbet     | 13.4     | 83.4 | 75.8   | 89.6  | 84.6 | 59.2 | 78.4 | 73.6  |
>
> We did not include SpikeBERT and SpikeGPT because they used different datasets except the SST-2. A comparison of these two models using SST-2 would be:
>
> | Model  | Size(MB) | Energy(mJ) | Acc  |
> | ---------- | -------- | ---------- | ---- |
> | SpikeGPT   | 216    | -   | 88.8 |
> | SpikeBERT  | -    | 28.54      | 85.4 |
> | **Sorbet** | 13.4     | 0.56       | 89.6 |
>
> Regarding ANN to SNN conversion methods, we performed the conversion after quantizing activations to 4 bits. This approach aligns with the mainstream ANN-to-SNN conversion techniques, such as QCFS [1] and [2]. The core idea behind [1] and [2] is to clip and quantize the activations to make the ANN model behave more like an SNN, thereby minimizing conversion loss. Our work follows this approach, incorporating advanced multi-step distillation inspired by [3] to obtain a quantized model with improved performance.
>
> [1] Bu T, Fang W, Ding J, et al. Optimal ANN-SNN Conversion for High-accuracy and Ultra-low-latency Spiking Neural Networks[C]. ICLR, 2023.
>
> [2] Shen G, Zhao D, Li T, et al. Are conventional snns really efficient? a perspective from network quantization[C]. CVPR, 2024.
>
> [3] Liu Z, Oguz B, Pappu A, et al. Bit: Robustly binarized multi-distilled transformer[J]. Neurips, 2022.
>
> ---
>
> > Q3: Results in Table 2
>
> In Table 2, we reported 2 results for SpikingBERT and SpikeLM. The results of their original paper are in lines 336 and 337. We noticed they further quantized their models to 1-bit ( SpikeLM reported in their original papers, while SpikingBERT reported separately in [4]). To make a fair comparison, we included quantized results in lines 338 and 339, denoted as 1-bit models.
>
> [4] Bal M, Jiang Y, Sengupta A. Exploring Extreme Quantization in Spiking Language Models[C]// ICONS, 2024.
>
> ---
>
> > Q4: Ablation for different timesteps
>
> The timestep used for all results reported is 16. We will make this explicit in the paper. We performed an ablation study for timesteps on SST-2 dataset:
>
> | Timestep | 2  | 4 | 8  | 16 |
> | ------ | ---- | ---- | ---- | ---- |
> | Accuracy | 78.7 | 87.4 | 89.3 | 90.4 |
>
> ---
>
> > Q5: Small typos and code accessibility
>
> Thank you for pointing out. We fix them and take your suggestion to replace Figure 3 with a table. The repository had expired, but we have extended it. The code is available now in the same link provided in the manuscript.
>
> ---
>
> > Q6: Use Sorbet as a multimodal LLM
>
> The components we have designed can be used on any model that uses the transformer mechanism. Therefore, it should be easily extendable to multimodal LLMs with customized training processes.
>
> ---
>
> > Q7: Calculation of energy saving
>
> We calculate our energy savings as:
> $$
> N_{\text{saving}} = \frac{E_{\text{BERT}}}{E_{\text{Sorbet}}} = \frac{15.21}{0.56} = 27.16
> $$
>
> We use 15.21mJ for FP16 BERT_base from SpikeLM (Xing et. al, 2024). For $E_\text{Sorbet}$, as in Appendix D.1:
>
> $$
> E_{\text{Sorbet}} = T \cdot r \cdot E_{\text{BERT}} \cdot \frac{E_{AC}}{E_{MAC}}.
> $$
> where $r$ is $0.13$ and $T$ is $16$. When using ${E_{MAC}} = 4.6pJ$, the $E_{\text{BERT}}$ should be FP32 BERT as 51.41mJ. Our operations are essential for SNNs but contribute less to energy savings, so we excluded them from the full model evaluation.

---

> > ### Comment · Reviewer_GqJG · 2025-04-01
> >
> > Appreciate the additional experiments and explanation.  I have raised the score.

---

> > > ### Author Response · Authors · 2025-04-02
> > >
> > > Thank you for reading our rebuttal and for updating the review!

---

### Decision · Program_Chairs · 2025-05-01

**Decision:**

Accept (poster)

**Comment:**

This paper proposes Sorbet, a spiking language transformer optimized for neuromorphic hardware. It incorporates a shifting-based softmax (PTsoftmax) and a BitShifting-based PowerNorm (BSPN) to replace the key operations softmax and layer normalization (LN) are difficult to implement on neuromorphic hardware. The proposed Sorbet achieves competitive performance with 27.16 times theoretical energy  savings compared to BERT.

After the rebuttal, three reviewers rate 3, 3, 4, respectively. All of the reviewers vote to accept. Reviewer GqJG thinks the proposed PTsoftmax and BSPN are interesting and promising. Reviewer MRPY thinks this work constructs an attention mechanism using a shift-based method while preserving a Softmax-like structure, making it a valuable contribution to the field. There are also some concerns remain. Reviewer GqJG and ck8m believe that it would be better if the authors deploy Sorbet on neuromorphic hardware. I think these reviews are justified. Although there are still some problems, I believe that this paper is worth accepting. Therefore, my recommendation is to accept this paper.